# Childhood Cancer Survivors Have Impaired Strain-Derived Myocardial Contractile Reserve by Dobutamine Stress Echocardiography

**DOI:** 10.3390/jcm12082782

**Published:** 2023-04-09

**Authors:** Olof Broberg, Ingrid Øra, Constance G. Weismann, Thomas Wiebe, Petru Liuba

**Affiliations:** 1Department of Pediatric Cardiology, Skane University Hospital, SE-221 85 Lund, Sweden; 2Department of Clinical Sciences, Pediatrics, Lund University, SE-221 85 Lund, Sweden; 3Department of Pediatric Oncology, Skane University Hospital, SE-221 85 Lund, Sweden; 4Department of Pediatric Cardiology and Pediatric Intensive Care, Ludwig-Maximilian University, 80539 Munich, Germany

**Keywords:** childhood cancer survivors, cardiomyopathy, myocardial strain imaging, anthracyclines, radiotherapy

## Abstract

Abnormal left ventricular contractile reserve (LVCR) is associated with adverse cardiac outcomes in different patient cohorts and might be useful in the detection of cardiomyopathy in childhood cancer survivors (CCS) after cardiotoxic treatment. The aim of this study was to evaluate LVCR by dobutamine stress echocardiography (DSE) combined with measures of myocardial strain in CCS previously treated with anthracyclines (AC). Fifty-three CCS (age 25.34 ± 2.44 years, 35 male) and 53 healthy controls (age 24.40 ± 2.40 years, 32 male) were included. Subjects were examined with echocardiography at rest, at low-dose (5 micrograms/kg/min), and at high-dose (40 micrograms/kg/min) dobutamine infusion. Left ventricular ejection fraction (LVEF) and global longitudinal strain (GLS), strain rate (GSR), and early diastolic strain rate (GEDSR) at different DSE phases were used as measures of LVCR. The mean follow-up time among CCS was 15.8 ± 5.8 years. GLS, GSR, and LVEF were lower at rest in CCS compared to controls (*p* ≤ 0.03). LVEF was within the normal range in CCS. ΔGLS, ΔGSR, and ΔGEDSR but not ΔLVEF were lower in CCS compared to controls after both low- (*p* ≤ 0.048) and high-dose dobutamine infusion (*p* ≤ 0.023). We conclude that strain measures during low-dose DSE detect impaired myocardial contractile reserve in young CCS treated with AC at 15-year follow-up. Thus, DSE may help identify asymptomatic CCS at risk for heart failure and allows for tailored follow-up accordingly.

## 1. Introduction

Cancer treatment regimens for different childhood cancer diagnoses are nowadays associated with survival rates exceeding 80% in the long term. Childhood cancer survivors (CCS) are thus an increasing population worldwide, with nearly 500,000 survivors being followed in Europe [1]. Cardiovascular diseases (CVD] amount to the most prevalent non-cancerous causes of death in CCS later in life [2,3]. In adult CCS, the relative risk for heart failure has been estimated to be 5.5 times higher compared to the general population [4], and the cumulative incidence of heart failure has been estimated to be as high as 10.6% 40 years after a childhood cancer diagnosis [5].

The reason for the increased risk of long-term heart failure in CCS appears to be due to anti-cancer treatment regimens involving chemotherapy and radiotherapy [6]. Cardiotoxic anthracyclines (AC) can cause both acute and chronic cardiac side effects, with a dose-dependent relationship. Despite the cumulative effect of AC, even very low cumulative doses can cause abnormalities in the ventricular function at follow-up [7]. Once clinically manifest, the prognosis of cancer treatment-related cardiac dysfunction (CTRCD) is poor [8].

The optimal echocardiographic measures of the progression of subclinical cardiac ventricular dysfunction to irreversible functional changes preceding clinically manifest heart failure are not yet established. Currently, the left ventricular ejection fraction (LVEF) is the mainstay parameter to evaluate cardiac function in CCS but is hampered by the inability to detect CTRCD at an early stage [9]. Left ventricular myocardial strain (global longitudinal strain, GLS) is a more sensitive marker of early subclinical LV systolic dysfunction. Abnormal strain has been observed in CCS at long-term follow-up despite normal LVEF [10] and has been shown to predict subsequent deteriorations of LVEF and exercise capacity [6,11].

Left ventricular contractile reserve (LVCR), as measured by changes in cardiac function during rest and during cardiac stress, has been of mounting interest in evaluating CTRCD to detect early cardiac dysfunction [12,13,14]. LVCR can be assessed by changes in LVEF during stress echocardiography, which is widely available and fairly inexpensive [13]. LVCR is also reduced in patients with the subtype of heart failure with preserved LVEF and diastolic dysfunction [15]. Strain imaging compared to LVEF might enable a better characterization of LVCR by directly interrogating the myocardium instead of the volumetric changes during the heart cycle [13,16,17].

Dobutamine, a synthetic catecholamine acting mainly on cardiac β_1_˗receptors, is the principal agent used for pharmacological stress testing of cardiac function in dilated cardiomyopathy [12,18]. Dobutamine stress echocardiography (DSE) is considered safe with minor dose-dependent side effects and does not require the participant to exercise, thereby enabling better conditions for image acquisition, vital for accurate strain imaging, than during conventional exercise stress echocardiography using a treadmill or a bicycle [18].

Previous studies of a strain-derived LVCR in CCS shortly after ending treatment (mean age at study 13–16 years) undergoing exercise stress echocardiography (ESE) with myocardial strain imaging showed no significant differences between CCS and controls [16,17]. We hypothesized that DSE in combination with myocardial strain imaging in young adult CCS later in life than previously studied would detect impaired LVCR because of the cumulative increase in heart failure in CCS with longer follow-up time. To investigate this, we examined CCS with DSE after being exposed to cardiotoxic AC with or without mediastinal radiotherapy.

## 2. Materials and Methods

We conducted a single-centre, prospective cohort study of myocardial strain imaging combined with DSE in young adults who were treated for childhood cancer at the Department of Paediatric Oncology of the Skåne University Hospital in Lund, Sweden. CCS were identified in the registry for childhood malignancies in southern Sweden [19]. Inclusion criteria: childhood cancer diagnosis under the age of 18, treatment with AC, survival more than 5 years after the disease remission, and a current age between 20–30 years. Exclusion criteria: a brain tumour diagnosis, previous cardiovascular disease (CVD) or any cardiovascular complication during cancer treatment, any chronic disease or syndrome, and pregnancy.

In total, 152 CCS met the eligibility criteria and received a written invitation to participate. If no answer was received, an additional invitation was sent. An equal number of healthy controls with similar sex and age were recruited by written announcements at the Skåne University Hospital area in Lund, Sweden, and these were examined in the exact same way and during the same time period as the CCS. Informed written consent was obtained from all study participants. The study protocol was approved by Lund University’s Regional Ethical Committee for Human Research (DNR 2013/205).

### 2.1. Clinical Data

All study participants completed a questionnaire [20] previously used in our institution covering current use of medications (cardiovascular medicines, statins), tobacco use (type, i.e., cigarettes or smokeless tobacco, dose, and frequency), and level of physical exercise (sports or gym-training, hours/week). Systolic and diastolic brachial blood pressure were measured in the supine position after 15 min of rest in the right arm using a calibrated wall-hung aneroid sphygmomanometer. Hypertension was defined as systolic blood pressure > 130 mmHg or diastolic blood pressure > 80 mmHg. Weight and height were measured (using a calibrated scale and a stadiometer). Obesity was defined as BMI > 30 kg/m^2^. To evaluate cardiotoxicity risk status, CCS were assigned to risk groups according to the 2015 guidelines published by the International Late Effects of Childhood Cancer Guideline Harmonization Group based on the cumulative AC-dose and chest radiation doses [21].

### 2.2. Dobutamine Stress Protocol and Echocardiography

All dobutamine stress echocardiographic (DSE) examinations followed a standardized protocol utilized at the Department of Cardiology, Skåne University Hospital, including three phases for echocardiographic evaluation; baseline, low phase (5 μg/kg/min dobutamine) and high phase (40 μg/kg/min dobutamine and up to 0.5 mg of atropine to reach target heart rate). The target heart rate for the peak phase was 220 −(0.85*age). DSE was terminated if psychological unrest, the occurrence of arrythmia, declining blood pressure, severe hypertension, or chest pain occurred. Participants were monitored by continuous electrocardiography (ECG) during the DSE. Blood pressure measurements were done at baseline and at each phase of dobutamine infusion.

Echocardiography was performed by a single investigator (OB). A cardiac ultrasound system (EPIQ-7, Philips Medical Systems, Andover, MA, USA) equipped with an X5-1 probe (Philips Medical Systems, Andover, MA, USA, frequency ranging between 1–5 MHz) was used. Echocardiography was done according to a standardized protocol, and image acquisition was performed according to the American Society of Echocardiography (ASE) guidelines [22]. All examinations were performed with the participant in the lateral decubitus position, but during DSE the participant could be put in another position for a better image acquisition. Time was taken to ensure optimal image quality during each phase. At the three different phases, image loops of eight ECG-gated cardiac cycles were acquired to ensure enough heart cycles of adequate quality for offline analysis. Left ventricular apical 2-, 3- and 4-chamber views were acquired. The image sector was focused on the left ventricle to achieve desired frame rates, and precaution was taken to optimize images for off-line strain analysis. Apical 2- and 4-chamber views were used for the calculation of left ventricular ejection fraction (LVEF) using the Simpson Bi-plane method of discs. LVEF was calculated for all DSE phases. Abnormal resting LVEF was defined as <52% for males and <54% for females [23].

During the baseline phase and at peak phase DSE, tissue doppler images were acquired for calculations of the mitral lateral and septal é- and ś-wave velocities. For right ventricular function, the tricuspid annular plane systolic displacement (TAPSE) was calculated using M-mode echocardiography.

For strain measures, the apical 2-, 3- and 4-chamber image loops acquired during the three phases were analysed offline using the TOMTEC 2D Cardiac Performance Analysis software (version 1.3.0.147, TOMTEC imaging systems, Unterschleissheim, Germany) using a semi-automatic algorithm. The stored cardiac cycles were inspected for breathing artifacts and dropouts. Three consequent cardiac cycles were chosen and analysed for each loop at each stage. End-diastolic and end-systolic time points were inspected to be correctly set by the software. The mean longitudinal values from 18 segments (six segments per each chamber view) for strain (S), strain rate (SR), and early diastolic strain rate (EDSR) were calculated and expressed as global longitudinal strain (GLS), global strain rate (GSR), and global early diastolic strain rate (GEDSR). Abnormal resting GLS was defined by values higher than –16% [24]. During DSE, left ventricular contractile reserve (LVCR) strain measures (GLS, GSR, GEDSR) were defined as abnormal if they were differing more than 2 SD of the control group.

Inter- and intra-observer variability for the ultrasound measurements were assessed in a subgroup of 23 studied subjects with two readers blinded to the status of the participants. Interobserver and intra-observer intraclass correlation coefficients (ICC) for all tissue doppler measurements and TAPSE were > 0.90. ICC for LVEF was > 0.84 for inter- and intra-observer ICC. For strain measurements (GLS, GSR and GEDSR) intra- and interobserver ICC were > 0.89 for baseline and low phase and at peak phase 0.91 for GEDSR, 0.86 for GLS, and 0.76 for GSR.

### 2.3. Statistical Analyses

Continuous variables were presented as median and range or as mean and standard deviation (SD) if the variable was normally distributed. Normal distribution was assessed by P-P plots of the residuals and histogram inspection. Categorical variables were presented as number (N) and frequency (%). A repeated measures ANOVA was performed to analyse differences in outcome variables between groups at the three phases, correcting for age and sex. Both means and estimated means were presented. Validity of the repeated measures ANOVA test was done to evaluate sphericity, and if this was not met a Greenhouse–Geisser coefficient > 0.75 was considered valid. ANOVA with age and sex as covariates was used to calculate differences between CCS and control subjects for numerical changes of these variables between phases. Means were presented. Simple linear regression was used to investigate cardiotoxic risk factors in CCS. Fisher’s exact test was used for dichotomous variables. *p*-values < 0.05 were considered statistically significant. Statistical analyses were performed in Statistical Package for Social Sciences, version 27 (IMB SPSS, version 27.0.0, Chicago, IL, USA).

## 3. Results

In total, 53 CCS and 53 age- and sex-matched controls were included. Their baseline characteristics are outlined in Table 1. The mean AC dose for all CCS was 211 mg/m^2^ (95% CI: 182.02–237.39). The median age at diagnosis was 7.73 years (range 0.75–17.72 years). The median follow-up time since cancer treatment completion was 16.41 years (range 6.05–26.91 years).

CCS were grouped according to the previously mentioned guidelines (Armenian, Hudson et al., 2015) (Table 2). Thus, there were four CCS with low-, 30 with moderate-, and 19 with high cardiotoxic risk according to current guidelines. In the low-risk group, there was one patient with Wilms tumour and three with Non-Hodgkin Lymphoma (NHL). In the moderate-risk group, the most common diagnosis was acute lymphoblastic leukaemia (ALL). Hodgkin’s disease (HD) was the most common diagnosis in the high-risk group.

The three NHL CCSs in the low-risk group had cumulative AC doses of 98–99 mg/m^2^. In the high-risk group, HD patients had a median AC dose of 160 mg/m^2^ (range; 157–259), while sarcoma, ALL, and AML patients had a median dose of 315.8 mg/m^2^ (range; 300–392), 370.3 mg/m^2^ (range; 308–446) and 422.0 mg/m^2^ (range; 315–446), respectively. Mediastinal radiotherapy was used in HD patients only. All CCS were free from any cardiac symptoms or diagnosis. No CCS were on cardiac medication or lipid-lowering medication at the time of the study.

### Cardiac Measurements at Rest and during DSE

Baseline cardiac measures for systolic and diastolic function at the three different DSE phases are outlined in Table 3, Table 4 and Table 5. Frame rates for apical 2-, 3- and 4-chamber views were median 64 (range, 58–86) frames/second at rest, at low-dose DSE 65 (range, 60–89) frames/second and 71 (range, 55–89) at high-dose DSE. LVEF and strain parameters were measurable at baseline and during the low phase in all CCS and controls. During the peak phase, strain images from three CCS and two controls were inadequate for analysis and two CCS had to interrupt the test because of irregular heart rate and one control due to anxiety. Therefore 50 (94.0%) controls and 48 (90.5%) CCS were included in the ANOVA for repeated measurements at the peak phase.

As shown in Table 3, CCS had a slightly higher resting heart rate and diastolic blood pressure (*p* > 0.048). CCS had less negative baseline GLS and GSR values compared with controls (*p* < 0.001). Four out of 53 CCS (7.5%, ns) had abnormal GLS (>–16%), and these had normal LVEF and were females with cumulative AC-doses of 222, 315, 422, and 446 mg/m^2^, respectively, with one patient with a history of ALL and three patients with a history of AML. No CCS had abnormal baseline GSR and one (1.8%, ns) had abnormal GEDSR.

LVEF and TAPSE were lower in CCS compared with controls (*p* < 0.05). Tissue Doppler lateral wall ś-wave velocity and septal and lateral é-wave velocities were lower in CCS compared with controls (*p* = 0.013 and *p* = 0.001, respectively).

During DSE at the low phase the heart rate was higher in CCS compared to controls (*p* = 0.003). During this phase, LVEF was lower (*p* ≤ 0.003), but the change (ΔLVEF) was not different in CCS compared to controls. CCS had lower GLS, GSR, and GEDSR (*p* ≤ 0.009) and a lower increase (ΔGLS, ΔGSR, and ΔGEDSR) from baseline (*p* ≤ 0.048 for all) compared with controls, as shown in Table 4.

Three out of the CCS with the abnormal GLS at baseline remained abnormal, and in total GLS was abnormal in nine CCS (17.0%, *p* = 0.003), for GSR in six CCS (11.3, ns and GEDSR in three CCS (5.6%, ns). Among the nine CCS with abnormal GLS during low-dose DSE, the cumulative AC-dose was 151–471 mg/m^2^. One of the CCS also had abnormal LVEF (1.8%, ns).

At the peak-phase [Table 5], the heartrate was higher in CCS than in controls (*p* = 0.007). GLS, GSR, and GEDSR were less negative in CCS compared to controls (*p* ≤ 0.001). The increase from baseline was lower in CCS for all strain measures compared with controls (*p* ≤ 0.048). LVEF was lower in CCS compared to controls (*p* ≤ 0.008), but the increase from baseline was not significantly different between the groups. At the peak-phase, three out of the CCS with the abnormal GLS at baseline remained abnormal, and in total GLS was abnormal in five CCS (10.4%, ns), for GSR in four CCS (8.3%, ns) and GEDSR in seven CCS (14.6%, *p* = 0.028).

Both septal and lateral systolic ś wave velocities were lower in CCS than in controls (*p* ≤ 0.007) and the increases in these from baseline were lower in CCS than in controls (*p* ≤ 0.019). Diastolic é wave velocities were similar between controls and CCS. TAPSE, and the change in TAPSE from baseline to peak phase, were lower in CCS compared to controls at the peak phase (*p* < 0.001).

As shown in Table 6, follow-up time after cancer treatment correlated weakly with GLS at rest (*p* = 0.047) and at the low phase (*p* = 0.004) but not at the peak phase. The cumulative AC dose was significantly correlated to GLS at baseline (0.011) and even more strongly at the low phase (*p* < 0.001) while more weakly at the peak phase (*p* = 0.023). There were no significant correlations to different cardiac measures after mediastinal radiotherapy.

For the cardiac risk group analysis, low- and moderate-risk groups were pooled and analysed against the high-risk group. The reason for this was that the low-risk group included only four CCS with a cumulative AC-dose close to 100 mg/m^2.^, which is the limit for moderate cardiotoxic risk. The cardiac risk group was, however, not significantly associated with GLS at any phase.

## 4. Discussion

In the current study, CCS with previous AC treatment LVCR were assessed at a mean follow-up time of 15.8 years using DSE. The main findings are: (1) 7.5% of CCSs had abnormal strain at rest, and this number increased to 17% at low-dose DSE. Moreover, augmentation of strain measures was impaired in CCSs when compared to controls. LVEF was not sensitive enough to detect these differences. (2) High-dose DSE does not seem to add further value in the evaluation of LVCR by strain measures, and the accuracy of myocardial strain at high heart rates is low, due to unphysiological loading conditions, technical limitations, and side effects.

### 4.1. Resting Cardiac Function

In the current study, CCS had lower strain measures (GLS and GSR) at rest than controls. This is in line with previously published data on long-term follow-up in CCS [24,25,26,27]. In the current study, four out of 53 CCS (7.5%) had clinically relevant reduced GLS values and all of these had normal LVEF values. Previous cross-sectional studies in CCS have shown reduced GLS early after cancer treatment [28,29]. In adult cancer survivors, a reduction in GLS of > 15% during cancer treatment is associated with later occurrence of overt cardiotoxicity including decline in LVEF and heart failure [30]. The current guidelines (2015) for the long-term follow-up for cardiotoxicity, however, do not include GLS because of the limited data on its clinical usefulness in long-term survivorship [21].

A recent longitudinal study by Pourier et al. (2020) of 41 survivors of childhood ALL followed with GLS showed that GLS continuously decreased with more than 10% from baseline in 54% of patients at > 5 years after treatment with preservation of LVEF [31]. Indeed, GLS appears to depend on follow-up duration; thus, cross-sectional studies with shorter follow-up may show subtle subclinical or no differences [29,32], whereas studies with a longer follow-up show abnormal GLS values compared to the normal population [24,27]. In prediction models for heart failure in CCS, the increased risk for heart failure follows the follow-up time [33]. In the current study (with a mean follow-up time of 15.8 years) GLS was significantly associated with follow-up time at baseline and more strongly associated with this during low-dose DSE.

The lack of pre- and during-treatment imaging suitable for GLS analysis could contribute to there still being little evidence linking reduced GLS to adverse outcomes in this population. However, given the large cumulative incidence of heart failure in CCS with a long latency time after cardiotoxic treatment [6], the mounting evidence that GLS is impaired in CCS, and that GLS precedes cardiotoxicity in adult cancer survivors, it is probably useful to include GLS in the long-term follow-up [34]. Further studies in this regard are warranted. Implementation of GLS measurements in CCS already before starting the anti-cancer treatment would enable clinicians to establish baseline measurements for each patient that might be useful in guiding clinical decisions such as increased surveillance and in predicting the risk for cardiovascular disease. In the general clinical perspective, there is also a need for standardized reference values and thresholds [35].

### 4.2. Myocardial Contractile Reserve

To our knowledge, four previous studies have used echocardiographic strain measures to characterize strain-based LVCR in CCS utilizing ESE [16,17,36,37] and none utilizing DSE. Kaneko et al. (2016) [36] described a lower LVCR (ΔGSR, ΔGEDSR) during peak exercise and found a significantly impaired response during peak exercise in CCS at a median age of 16 years (range 8–19 years) compared with controls. Noteworthy is that this cohort consisted initially of 33 CCS, but 11 had inadequate image quality and were excluded. The authors used a basal parasternal short-axis view to calculate strain measures and therefore analysed only a few left ventricular segments to establish mean values.

Two other studies, Ryerson et al. (2015) and Cifra et al. (2018) [16,17] found no significant differences at peak exercise. Ryerson et al. used tissue-Doppler-derived strain measures, enabling higher frame rates, but due to tachycardia and reduced image quality at peak exercise they could not analyse strain and strain rate. Cifra et al. used a semi-supine bicycle for exercise, which allowed them to acquire images with better quality during exercise; of the 100 studied CCS, all had adequate image quality to analyse strain measures, although they only used the apical 4-chamber view to analyse strain. In the study of von Scheidt et al. (2022) [37], 77 CCS were studied, and no differences between CCS and controls regarding GLS and GSR were found. However, there was a trend for an increased abnormal GLS and GSR with increasing exercise. The above studies [16,17,37] studied CCS with a relatively short follow-up of 8–10 years. Ryerson et al. (2015) and Cifra et al. (2018) concluded that CCS improve cardiac function through an increase of exercise, similar to healthy controls [16,17].

We show, in the current study, a decreased LVCR in this CCS cohort by strain measures by using dobutamine as a stressor. At baseline, 7.5% of CCS had abnormal GLS, and with low-dose DSE (5 μg/kg/min) this number increased to 17% as defined by a cut-off value of 2 SD of the control group. These findings are important, as symptoms of heart failure develop during exercise when there is a demand for increased cardiac output. The differences in the current study compared to the above-mentioned studies [16,17,37] are probably due to a longer follow-up time—which Von Scheidt et al. [37] also concluded in their work. Longer follow-up time and younger age at diagnosis have been established as correlates for cardiotoxicity in CCS [6,9], and therefore this aspect must be considered when evaluating LVCR in CCS. Further, in the current study, a full left ventricular tri-plane analysis of GLS was performed. It is possible that despite the idea that cardiotoxicity caused by AC is a global phenomenon, some segments can be more affected than others. By using only one view, two thirds of left ventricular segments are possibly missed, and this could be particularly important in patients after mediastinal radiotherapy when parts of the heart and large vessels are in the radiotherapy field [38].

### 4.3. Difficulties with High-Dose DSE

With an increased DOB dose, we observed no additive information comparing mean values between CCS and controls. The lower response at high-dose DSE (40 μg/kg/min) of both LVEF and GLS is due to lower preload because of unphysiologically low systemic venous return, causing lower cardiac output compared to ESE [39]. Additionally, at high-dose DSE there were side effects and significant discomfort for the participants. Therefore, we suggest that strain measures should not be used with high-dose DSE.

### 4.4. TAPSE and Tissue Doppler

In the current study, TAPSE was lower at rest and during high-dose DSE in CCS compared to controls, indicating that cardiotoxicity is also present in the right ventricle. Tissue Doppler systolic ś-wave velocities of the lateral wall were lower at rest in CCS compared to controls, but septal velocities did not differ. However, at high-dose DSE both septal and lateral wall ś-wave velocities were lower in CCS compared to controls, suggesting that an impairment in LVCR in CCS can also be measured by this method. TAPSE and tissue-Doppler measures are fast and simple to acquire and might be better in a clinical setting to evaluate LVCR in CCS. Unfortunately, we did not measure tissue Doppler with low-dose DSE, which would have enabled a comparison with strain measures.

### 4.5. Underlying Mechanisms

The mechanism for AC cardiotoxicity includes the formation of reactive oxygen species and inhibition of cardiac mitochondrial topoisomerase-2β, resulting in cardiac myocyte loss and fibrosis [40]. CCS after AC-treatment have been shown by cardiac magnetic resonance imaging to have diffuse myocardial fibrosis, and this might explain the subclinical cardiac abnormalities in CCS [41]. AC exhibit cardiotoxicity starting mainly in the subendocardial layer with longitudinally oriented fibres; thus, longitudinal strain measures might be sensitive in detecting AC cardiotoxicity [25]. In the current study, GLS correlated with the cumulative AC-dose, suggesting that these observations are due to AC-induced myocardial fibrosis and loss of myocytes.

The development of long-term complications including heart failure and other cardiovascular late effects in CCS is most likely multifactorial. Modifiable cardiovascular risk factors are prevalent in CCS and act in synergy with cardiotoxicity due to anti-cancer treatments in further increasing the risk for cardiovascular disease [6]. We have previously shown unfavourable lipid and apolipoprotein profiles in this CCS cohort [20]. Lipshultz et al. recently showed that even among CCS with similar or better cardiometabolic and lifestyle profiles compared with population-matched controls, there remains a higher risk for future clinically significant cardiovascular disease including heart failure [42]. GLS has been shown to correlate to traditional cardiovascular risk factors [43]. Clearly, aggressive management of modifiable cardiovascular risk factors is recommended to optimize the cardiac health in CCS [44].

### 4.6. Strengths

The strengths of the here presented study are (1), the young age of the study cohort (mean age 24.40 years), (2), no chronic diseases, cardiac symptoms or medications that could have confounded the results, (3), the use of tri-plane measurements to characterize the whole left ventricle during DS, and (4), the use of dobutamine as stressor instead of exercise for improved image acquisition.

### 4.7. Study Limitations

DSE has the advantage of better image acquisition compared to ESE, since ESE causes breathing and movement artifacts. However, image acquisition during high-dose DSE was feasible—it was not easily accomplished in several participants because high doses of dobutamine caused discomfort in some patients who needed to move or take deep breaths, making image acquisition harder and more time-consuming. Another limitation of this study is that we did not measure other markers for decreased cardiac reserve such as VO2 uptake, exercise capacity, and pulmonary function, which would further have characterized LVCR. The number of CCS included in this study was only 53, so a type 2 error may have occurred. A bigger cohort with additional inclusion of patients with a cardiovascular diagnosis might have given an association with adverse outcomes.

In conclusion, low-dose DSE (5 μg/kg/min) in combination with GLS may be a useful method for assessing the left ventricular myocardial contractile reserve (LVCR) in CCS. We suggest that low-dose DSE combined with GLS should further be evaluated as a possible investigation of routine echocardiographic follow-up of CCS in young adulthood for better risk stratification of CCS, at least in the high-risk group.

## Figures and Tables

**Table 1 jcm-12-02782-t001:** Baseline Characteristics of CCS and Healthy Controls.

Variable	Controls, n = 53, Mean (SD)	CCS, n = 53, Mean (SD)	*p*-Value
Sex (M/F)	35/18	32/21	ns
Age (years)	25.34 (2.44)	24.40 (2.40)	ns
Height (cm)	179.11 (8.73)	174.83 (10.35)	0.023
Weight (kg)	79.25 (15.32)	74.84 (14.10)	ns
BMI, (kg/m^2^)	24.57 (3.79)	24.40 (3.53)	ns
BSA (m^2^)	1.98 (0.23)	1.91 (0.22)	ns
Obese (n, %)	5 (9.43)	4 (7.54)	ns
Ever smoke (n, %)	9 (16.98)	11 (20.75)	ns
Exercise (h/week)	4.44 (2.87)	4.94 (8.32)	ns
Cumulative AC-dose (mg/m^2^)		211.71 (93.17)	
Age at diagnosis (years)		8.40 (5.57)	
Follow up time (years)		15.78 (5.76)	
Mediastinal RT (n, %)		10 (18.87)	

Abbreviations: BMI, body mass index; BSA, body surface area; AC, anthracycline; RT, radiotherapy; ns, not significant.

**Table 2 jcm-12-02782-t002:** CCS Characteristics according to Cardiomyopathy Risk Group. Adapted with permission from [21], copyright 2023, OB.

	Low Risk, n = 4	Moderate Risk, n = 30	High Risk, n = 19
Diagnosis (n, %)			
Acute lymphoblastic leukaemia (ALL)	0 (0.0)	18 (60.0)	2 (10.5)
Acute myeloid lymphoma (AML)	0 (0.0)	0 (0.0)	3 (15.8)
Hodgkin’s disease (HD)	0 (0.0)	3 (10.3)	8 (42.1)
non-Hodgkin’s disease (non-HD)	3 (75.0)	4 (13.3)	1 (5.3)
Sarcoma	0 (0.0)	2 (6.6)	4 (21.1)
Wilms’ tumour	1 (25.0)	3 (10.0)	1 (5.3)
Cumulative AC ((mg/m^2^), median (range))	98 (50–99)	195.9 (101–248)	300 (157–471)
Age at diagnosis ((years), median, range)	9.6 (2.8–16.4)	5.1 (1.1–17.7)	10.2 (0.8–16.7)
Follow-up time ((years), median, range)	13.7 (6.1–20.8)	17.4 (6.1–24.1)	13.6 (6.3–26.9)
Mediastinal radiotherapy (n, %)	0 (0.0)	1 (3.4)	9 (45.0)

Abbreviations: AC, anthracyclines; Cardiomyopathy risk group according to International Late Effects of Childhood Cancer guideline Harmonization Group [21]: Low risk–cumulative AC-dose < 100 mg/m^2^; Moderate risk–cumulative AC- ≥ 100 < 250 mg/m^2^ or mediastinal radiotherapy ≥ 15 < 35 GY, high risk, cumulative AC-dose ≥ 250 mg/m^2^ OR mediastinal radiotherapy ≥35 GY OR cumulative AC-dose ≥ 100 mg/m^2^ and mediastinal radiotherapy ≥ 10 GY.

**Table 3 jcm-12-02782-t003:** Means and estimated means of Baseline cardiac parameters in CCS exposed to cardiotoxic treatments compared to in healthy controls.

Variable, Mean (SD)	Controls	CCS	*p*-Value
Heart Rate (beats/min)	59.32 (8.36)	63.72 (8.48)	0.014
Systolic Blood Pressure (mmHg)	116.96 (14.34)	118.70 (11.21)	Ns
Diastolic Blood Pressure (mmHg)	73.42 (5.63)	76.87 (10.78)	0.048
SBP LVEF (%)	60.79 (5.11)	58.09 (5.16)	0.035
é lateral wall (cm/s)	17.04 (3.91)	14.52 (2.29)	<0.001
é septum (cm/s)	12.26 (1.65)	11.05 (1.97)	0.001
ś lateral wall (cm/s)	9.73 (1.83)	8.70 (2.16)	0.009
ś septum (cm/s)	8.22 (1.18)	8.15 (1.17)	Ns
TAPSE (mm)	27.80 (5.13)	25.34 (4.51)	0.025
GLS (%)	−20.35 (2.01)	−18.81 (2.17)	0.012
GSR (1/s)	−0.97 (0.20)	−0.89 (0.16)	0.024
GEDSR (1/s)	1.22 (0.33)	1.11 (0.25)	ns

Abbreviations: SBP, Simpson Bi-Plane: TAPSE, tricuspid valve annular plane systolic excursion, GLS, global longitudinal strain, GSR, global longitudinal strain rate, GEDRS, global longitudinal early diastolic strain rate. Differences between means were analysed by ANCOVA using age and sex as covariates.

**Table 4 jcm-12-02782-t004:** Cardiac measurements during DSE at low dobutamine dose.

Variable	Controls	CCS	*p*-Value
Heart Rate (beats/min)	66.34 (10.41)	74.25 (15.07)	0.003 *
ΔHeart rate (beats/min)	7.05 (8.52)	10.18 (13.87)	ns
SBP (mmHg)	127.56 (17.05)	126.34 (12.04)	ns
ΔSBP (mmHg)	10.28 (16.75)	7.64 (13.62)	ns
DBP (mmHg)	84.17 (12.45)	86.64 (11.16)	ns
ΔDPB (mmHg)	10.55 (11.80)	9.77 (14.02)	ns
LVEF (%)	71.23 (5.07)	68.20 (5.35)	0.003
ΔLVEF (%)	10.27 (5.23)	9.79 (5.19)	ns
GLS (%)	−25.44 (2.68)	−22.65 (2.75)	<0.001
ΔGLS (%)	−5.13 (2.40)	−3.86 (2.27)	0.011
GSR (1/s)	−1.59 (0.28)	−1.38 (0.28)	0.004
ΔGSR (1/s)	−0.63 (0.24)	−0.49 (0.29)	0.040
GEDSR (1/s)	1.75 (0.39)	1.50 (0.32)	0.009
ΔGEDSR (1/s)	0.55 (0.34)	0.39 (0.31)	0.034

Abbreviations: SBP, Simpson Bi-Plane, TAPSE, tricuspid valve annular plane systolic excursion, GLS, global longitudinal strain, GSR, global longitudinal strain rate, GEDSR, global longitudinal early diastolic strain rate. Differences between means were analysed by Repeated measures ANOVA correcting for age and sex. * Multivariate test not significant.

**Table 5 jcm-12-02782-t005:** Cardiac measures during DSE at high dobutamine dose.

Variable	Controls	CCS	*p*-Value
Heart Rate (beats/min)	156.35 (8.12)	160.47 (8.76)	0.007
ΔHeart rate (beats/min)	97.13 (9.34)	96.9 (12.67)	ns
Systolic Blood Pressure (mmHg)	168.48 (26.00)	163.34 (26.16)	ns
ΔSystolic Blood Pressure (mmHg)	51.31 (25.74)	44.64 (27.71)	ns
Diastolic Blood Pressure (mmHg)	91.92 (15.87)	90.68 (14.70)	ns
ΔDiastolic Blood Pressure (mmHg)	18.42 (15.78)	13.81 (16.66)	ns
é lateral (cm/s)	18.41 (4.60)	17.54 (3.63)	ns
Δé lateral (cm/s)	1.37 (5.66)	3.05 (3.60)	ns
é septum (cm/s)	16.50 (3.53)	16.04 (3.29)	ns
Δé septum (cm/s)	4.23 (3.66)	5.00 (3.63)	ns
ś lateral wall (cm/s)	20.43 (4.92)	16.66 (3.76)	<0.001
Δ’s lateral wall	10.70 (4.82)	7.92 (3.66)	0.003
ś septum (cm/s)	17.44 (3.16)	15.77 (2.55)	0.007
Δ’s septum	9.23 (3.39)	7.59 (3.12)	0.019
TAPSE (mm)	27.91 (5.31)	21.67 (5.08)	<0.001
ΔTAPSE (mm)	0.11 (5.16)	−3.88 (5.74)	<0.001
LVEF (%)	67.95 (6.48)	66.94 (4.89)	0.018
ΔLVEF (%)	7.09 (6.71)	6.60 (6.34)	ns
GLS (%)	−22.89 (3.23)	−19.94 (2.39)	<0.001
ΔGLS (%)	−2.46 (3.42)	−1.17 (2.93)	0.048
GSR (1/s)	−2.24 (0.41)	−1.92 (0.39)	0.001
ΔGSR (1/s)	−1.27 (0.44)	−1.04 (0.37)	0.023
GEDSR (1/s)	2.19 (0.41)	1.82 (0.46)	<0.001
ΔGEDSR (1/s)	0.98 (0.42)	0.71 (0.46)	0.005

Abbreviations: SBP, Simpson Bi-Plane; TAPSE, tricuspid valve annular plane systolic excursion; GL, global longitudinal strain; GSR, global longitudinal strain rate; GEDRS, global longitudinal early diastolic strain rate. Differences between means were analysed by Repeated measures ANOVA correcting for age and sex.

**Table 6 jcm-12-02782-t006:** Univariate Linear Regression in CCS of GLS during DSE as the dependent variable.

	n	B (95%CI)	β	t	*p*-Value
Dependent variable: GLS (%) at baseline	53				
Age (years)		0.08 (−0.16–0.31)	0.09	0.64	ns
Sex		0.20 (−0.97–1.37)	0.05	0.35	ns
female	21				
male	32				
Follow-up time (years)		0.10 (0.001–0.19)	0.27	2.03	0.047
Mediastinal radiotherapy (y/n)	10	−0.96 (−2.40–0.48)	−0.18	−1.33	ns.
Cumulative dose AC (mg/m^2^)		0.01 (0.002–0.01)	0.35	2.64	0.011
Cardiac Risk-Group		0.85 (−0.32–2.02)	0.20	1.46	ns
low- and moderate risk group	34/53				
high-risk	19/53				
Dependent variable: GLS (%) at low dose DSE	53				
Age (years)		0.03 (−0.29–0.35)	0.03	0.19	ns
Sex		0.31 (−1.24–1.27)	0.06	0.41	ns
female (n)	21				
male (n)	32				
Follow-up time (y)		0.18 (0.06–0.31)	0.39	3.00	0.004
Mediastinal radiotherapy (y/n)	10	−1.80 (−3.67–0.08)	−0.26	−1.93	ns
Cumulative dose AC (mg/m^2^)		0.02 (0.10–0.024)	0.59	5.11	<0.001
Cardiac risk group		1.16 (−0.40–2.71)	0.21	1.50	ns
low- and moderate risk group	34/53				
high-risk	19/53				
Dependent variable: GLS (%) at peak phase	48				
Age (years)		−0.11 (−0.39–0.18)	−0.11	−0.76	ns
Sex		−1.11 (−2.51–0.29)	−0.23	−1.60	ns
female	29				ns
male	19				ns
Follow-up time (years)		0.08 (−0.04–0.20)	0.20	1.42	ns
Mediastinal radiation (y/n)		0.30 (−1.95–1.81)	0.01	0.03	ns
Cumulative AC-dose (mg/m^2^)	53	0.01 (0.001–0.02)	0.33	2.36	0.023
Cardiac risk-group		1.23 (−0.19–2.65)	0.25	1.74	ns
low- and moderate risk group	22				
high-risk	10				

Abbreviations: GLS, global longitudinal strain, AC, anthracycline, TAPSE, tricuspid valve annular plane systolic excursion, BMI body mass index.

## Data Availability

The data presented in this study are available upon request from the corresponding author.

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
