# Peer review of "Childhood Cancer Survivors Have Impaired Strain-Derived Myocardial Contractile Reserve by Dobutamine Stress Echocardiography"

_jcm, 2023, doi:10.3390/jcm12082782_

Round 1
Reviewer 1 Report
The Authors report on subclinical echocardiographic abnormalities in childhood cancer survivors. The topic is of interest, though largely assessed in literature.
I have two main comments regarding Discussion, which needs to be implemented. Two open questions remain regarding the usefulness of GLS and DSE in cancer survivors, that somehow limit their use in clinical practice. These aspects need to be addressed largely in discussion.
1) What is the real clinical significance of reduced GLS in these patients? There are no strong data linking reduced GLS to adverse outcomes. Thus, what is the utility of performing such tests and such CV risk stratification. This study lacks of a significant number of participants and of follow-up information to answer these questions.
2) It has been shown, on the contrary, that performing basic CV primary prevention strategies (i.e., control of CV risk factors) is effective in reducing the risk of HF in these patients. This appears to be a much more easy and effective method. Please discuss
3) Even chosing to use GLS as a measure to stratify CV risk in cancer survivors, what would be the best time to do so? This is very important, as time from cancer appears to be the only significant predictor of reduced GLS across all risk categories in cancer survivors.
Moreover, some other minor comments:
- you cannot say that you have a sex and age matched control group if the numbers of males/females are not the same
- if you report ranges (i.e., for times), then you have to use median and not mean values.
Reviewer 2 Report
This is very interesting study to evaluate myocardial contractile reserve by dobutamine stress echo in childhood cancer survivor (CCS).
Although the result of the study demonstrated that the DSE can differentiate myocardial contractile reserve of CCS group compared to control group, all echo parameters (baseline and during DSE) were in normal range. The changes of echo parameters were from normal ranges to normal ranges.
Which suggests that the myocardial contractile reseve of the CCS group may be impaired compared to control group, but it does not mean that the CCS group has cancer therapy related cardiac dysfunction.
I think it would be better to re-describe the conclusion.
"low-dose DSE (5µg/kg/min) in combination GLS is a useful method for 371 assessing CTRCD => myocardial constractile reserve in CCS."
In conclusion paragraph (line number 372), the CTRCD is not proper term, it should be changed to myocardial contratile reserve.
The definition of CTRCD should be considered.
European Heart Journal, Volume 43, Issue 41, 1 November 2022, Pages 4229–4361, https://doi.org/10.1093/eurheartj/ehac244
Round 2
Reviewer 1 Report
Authors ameliorated the paper after revision